# Plasma electrophoresis profiles of Blanding's turtles (*Emydoidea blandingii*) and influences of month, age, sex, health status, and location

Kirsten E. Andersson[1]*, Laura Adamovicz[1,2], Lauren E. Mumm[1], Samantha E. Bradley[1], John M. Winter[1], Gary Glowacki[3], Carolyn Cray[4], Matthew C. Allender[1]*

1 Wildlife Epidemiology Laboratory, College of Veterinary Medicine, University of Illinois, Urbana, IL, United States of America, 2 Veterinary Diagnostic Laboratory, College of Veterinary Medicine, University of Illinois, Urbana, IL, United States of America, 3 Lake County Forest Preserve District, Libertyville, IL, United States of America, 4 Department of Pathology & Laboratory Medicine, School of Medicine, University of Miami, Miami, FL, United States of America

* kkeandersson@gmail.com (KEA); mcallend@illinois.edu (MCA)

## Abstract

Baseline plasma electrophoresis profiles (EPH) are important components of overall health and may aid in the conservation and captive management of species. The aim of this study was to establish plasma protein fractions for free-ranging Blanding's turtles (*Emydoidea blandingii*) and evaluate differences due to age class (adult vs. sub-adult vs. juvenile), sex (male, female, or unknown), year (2018 vs. 2019), month (May vs. June vs. July), health status, and geographical location (managed vs. unmanaged sites). Blood samples were obtained from 156 Blanding's turtles in the summer of 2018 and 129 in 2019 at two adjacent sites in Illinois. Results of the multivariate analysis demonstrated that age class, sex, year, month, health status, and geographical location all contributed to the variation observed in free-ranging populations. Adult females had the highest concentration of many protein fractions, likely associated with reproductive activity. Juveniles had lower protein concentrations. Temperature and rainfall differences between years impacted concentrations between 2018 and 2019, while May and June of both years saw higher levels in some protein fractions likely due to peak breeding and nesting season. Individuals with evidence of trauma or disease also showed increased plasma protein fractions when compared to those that were considered healthy. The two sites showed a wide/large variation over the two years. All of these factors emphasize the importance of considering multiple demographic or environmental factors when interpreting the EPH fractions. Establishing ranges for these analytes will allow investigation into disease prevalence and other environmental factors impacting this endangered species.

**Data Availability Statement:** All relevant data are within the manuscript and its Supporting Information files.

## Introduction

Climate change, habitat destruction, and disease are potentially affecting the sustainability and conservation of species [1]. Wildlife health surveillance has become a critical component to

**Funding:** Funding was provided by a private charitable foundation, Crown Family Philanthropies (https://crownfamilyphilanthropies.org/). They do not wish to have any further information disclosed. The funders had no role in study design, data collection and analysis, decision to publish, or preparation of the manuscript.

**Competing interests:** The authors have declared that no competing interests exist.

maintain imperiled populations [1, 2], but determining health is complex and involves several modalities, including physical examination, clinical pathology, pathogen presence, and contaminant exposure [3–6]. Protein electrophoresis (EPH), a component of clinical pathology investigation, has become commonly used in wildlife studies [7]. Reference interval data in several reptilian species exist and have proven to be valuable in understanding how they handle stress and disease [4, 7–16].

Plasma proteins are key players in the body's innate immune response, and fluctuations in concentrations serve to indicate the presence of inflammation, infection, neoplasia, stress, or trauma [17]. In head-started red-bellied cooters (*Pseudemys rubriventris*) and captive-reared loggerhead sea turtles (*Caretta caretta*), variations in protein fractions were associated with differences in age, diets, immune stimulation, and reproductive stage [18, 19], indicating baseline differences exist and responses to changes in demographic and environmental variation can be measured.

The Blanding's turtle is a semi-aquatic, long-lived species of turtle experiencing population declines over much of its range in southern and central Canada and northern United States. Individual turtles can live up to 80 years of age in the wild [20], but urban development, road mortality events, climate change, illegal poaching, and disease remain the most common threats to sustainability [21]. The Chiwaukee Prairie–Illinois Beach Lake Plain (Lake Plain) contains a population of Blanding's turtles, in which active conservation efforts are aimed at improving the long-term viability in northeastern Illinois and southeastern Wisconsin [22]. In the summer of 2015, conservation efforts incorporated physical exam and health assessment data to aid in a greater understanding of the biological threats these animals face.

The objective of this study was to establish baseline plasma protein fractions for free-ranging Blanding's turtles and to determine differences between age classes (adult vs. sub-adult vs. juvenile), sex (male vs. female), years (2018 vs. 2019), months (May vs. June vs. July), health status, and geographical location. It was hypothesized that total protein levels would be higher in adults than subadults and juveniles, higher in females than males, higher in May than in June and July, and higher in unhealthy turtles than healthy turtles.

## Materials and methods

### Study sites

Blanding's turtles were sampled from three sites within the Lake Plain including Spring Bluff-Chiwaukee Prairie (SBCP), the managed site, and the North and South units of Illinois Beach State Park (IBSP), the unmanaged sites. SBCP consists of approximately 535 acres of high-quality coastal dune and swale habitat along the coast of Lake Michigan in Illinois and Wisconsin, whereas IBSP consist of 4,160 acres of dune, prairie, oak-savannah, and wetland habitats along 6.5 miles of coastline [22]. Management of mesopredators is also more robust in SBCP than IBSP, with studies focusing on camera trap surveillance for predator presence throughout the year as well as nest predation rates [23].

### Capture methods

Turtles were captured with the aid of radiotelemetry, hoop net trap, or incidentally by hand. Radiotelemetry is a three-part system using a radio transmitter, a radio antenna, and a radio receiver. The transmitter is attached to the turtle's shell, which transmits a signal to the antenna and correlates to a beeping produced by the receiver that gets louder as the animal gets closer [23, 24]. Hoop traps were placed in marsh waters and areas that were characteristic of Blanding's habitat or locations near previous Blanding's turtle capture sites. These traps were checked every 24 hours and remained in the same location for up to five days. Telemetry,

traps, and incidental captures were used in both field sites. Many turtles, especially those equipped with a radio transmitter, were sampled up to two times per summer.

## Physical examination and sample collection

Each turtle was assigned a permanent ID, marking the shell with a notch code unless previously marked as well as inserting a pit tag (microchip) under the skin, and mass, sex, and age class were recorded. Sex was classified as male, female, or unknown. Sex of head-started turtles, which are clutches deposited in captivity from free-range females, was known due to established incubation temperatures, as males were incubated at 26.5˚C and females at 31.0˚C [23, 25]. The sex of adults was determined based on plastron concavity [26]. The sex of most sub-adults and juveniles was estimated based on position of cloacal opening [27]. Wild-born individuals were classified as unknown sex when a confident determination could not be made. Age class was characterized as juvenile (<250 grams), sub-adult (250–750 grams), or adult (>750 grams). Blanding's turtles were deemed sexually mature at 750 grams and over by the Lake County Forest Preserve District (LCFPD) based on the lightest fertile female noted. This methodology is based on a previously published study by Mumm, et al. [24]. Body fat percentage (FP) was calculated using a published calculation from the relationship of carapace length and mass [28]. Physical examinations were performed noting visual appearance of the eyes, nose, oral cavity, ears, legs, digits, shell, integument, and cloaca. Gravidity was assessed using digital palpation of the prefemoral fossa. For the purpose of statistical analysis, females were classified as gravid if they had palpable eggs or if they had nested within one week of sampling. Nesting was determined either by observation of nesting behavior or the lack of palpable eggs after having previously been confirmed gravid. Turtles were classified as either "apparently healthy" or "unhealthy" based on the presence of clinically significant physical exam abnormalities, including open fractures or wounds; ocular, oral or nasal discharge; depressed mentation; missing nails, digits, or appendages; and evidence carapace/plastron damage.

Whole blood was collected from the sub-carapacial sinus via 22-gauge or 25-gauge needle, subject to the size of the individual. No more than 0.6% of body weight of whole blood was drawn and placed into lithium-heparinized plasma separator tubes. Blood samples were placed on ice in a cooler for one to five hours depending on time of collection until returning to the lab each afternoon. Total protein (TP) was estimated using refractometry. Whole blood samples were centrifuged at 4,185 g for 10 minutes, stored at -20˚C for one to four months, and shipped on dry ice to the University of Miami at the end of the field season. All individuals were released at coordinates of capture. All animal sampling was permitted by the following organizations: Department of Natural Resources (IDNR) (Scientific Collectors Permits (SCP): NH17.5065, NH18.5065, and IDNR Endangered and Threatened permits: SBT-16-062, 1199, 14–046, and 1042), the Wisconsin Department of Natural Resources (WIDNR) (SCP: SCP-SOD-004-2013 and WIDNR Scientific Research License: SRLN-18-026), and the University of Illinois Institutional Animal Care and Use Committee (Protocols: 18000 and 18165).

## Protein electrophoresis

Plasma samples were analyzed according to the procedure provided by the Helena SPIFE 3000 system with the use of Split Beta gels (Helena Laboratories, Inc., Beaumont, Texas 77707, USA). Results were produced after gel scanning and analysis by Helena software. Fraction delimits were placed as previously demonstrated for other reptiles [10]. Plasma protein fractions were divided into the following six fractions: a fraction migrating in the prealbumin region ("prealbumin"), albumin, alpha 1 globulins, alpha 2 globulins, beta globulins, and gamma globulins (Fig 1). Percentages for each fraction were determined by this software,

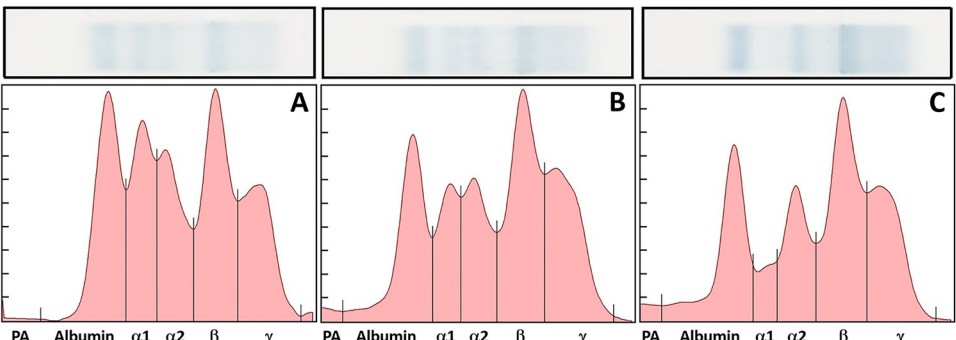

**Fig 1. Electrophoretogram comparison of plasma protein fractions.** A visual representation of the plasma protein fraction distribution amongst three representatives from each age classes. The top panel shows the protein as they appeared on the gel following electrophoresis, and the bottom panel shows a graphic representation of the concentration differences. Juvenile (A), sub-adult (B), and adult (C).

which gave the relative value, and absolute values (g/L) for each fraction were obtained by multiplying the percentage by the TP concentration. The albumin:globulin ratio (A:G) was calculated by dividing the sum of albumin and prealbumin by the sum of the globulin fractions.

## Statistical analysis

All statistical analyses were conducted in R version 3.6.3 [29] at an alpha level of 0.05, unless otherwise specified. Data distributions were assessed using histograms and the Shapiro-Wilk test and transformation was pursued, if necessary, to meet modeling assumptions.

Protein electrophoresis fractions were modeled using general linear mixed models with the lme4 and lmerTest packages [30, 31]. Fixed effects included spatiotemporal variables (year, month, location), demographic variables (sex, age class, gravidity), and health variables (body condition, presence/absence of physical exam abnormalities), while turtle ID was included as a random effect. Post-hoc testing was performed using the lsmeans package with a Tukey correction to control for multiple statistical tests [32]. Fixed effects with univariable p-values $< 0.15$ were considered in multivariable models testing specific biological hypotheses about the effects of spatiotemporal, demographic, and health variables on EPH values. Variance inflation factors were evaluated for multivariable models to identify and exclude highly collinear (VIF $> 10$) predictor variables (function vif, package car) [33]. Candidate model sets were constructed for each EPH fraction and ranked using an information-theoretic approach [34]. Figures were constructed using the ggeffects package [35]. Spatial clustering of EPH fractions was modeled in ArcGIS version 10.6 using hot spot analysis with the Gettis-Ord Gi* statistic from the spatial statistics toolbox. Hot spots are areas with higher plasma protein concentrations, while cold spots are areas with lower plasma protein concentrations.

Coefficients of variation were determined for each EPH fraction using data from apparently healthy turtles evaluated at multiple time points ($CV_I$) and only a single time point ($CV_G$). The index of individuality (II) was calculated as $CV_I / CV_G$ and was used to infer the need for subject-based vs. population-based reference intervals [36, 37]. When the II is $< 0.6$, subject-based reference intervals was used, while an II $> 1.4$ population-based reference intervals were created (Harris, 1974). When the II was between 0.6 and 1.4, population-based reference intervals were used [37, 38]. Reference change values (RCV) were calculated for each EPH analyte using a published formula [36, 37].

Population-based reference intervals were also determined for each analyte using the non-parametric method (referenceIntervals package) [39], according to American Society for

Veterinary Clinical Pathology guidelines [36]. Outliers were identified and excluded using Horn's method [40]. Ninety percent confidence intervals were generated around the limits of each reference interval using nonparametric bootstrapping with 5000 replicates. The population-based reference interval dataset included only turtles sampled once, and a randomly-selected single time point (https://www.random.org/) from serially-sampled individuals.

## Results

Two hundred and eighty-five samples were collected from 215 individual turtles. Fifty animals were sampled two times approximately one year apart, and ten animals were sampled three times–twice within the 2018 active season and a third time in 2019. One hundred fifty-six samples were obtained in 2018, while 129 were collected in 2019. One hundred seventy-eight samples were collected at SBCP, 51 were from IBSP North Unit, 54 were from IBSP South Unit, and two individuals did not have location data recorded. Samples were collected in May (N = 129), June (N = 120), and July (N = 36) from 171 adults, 83 sub-adults, and 31 juveniles. Sex distribution included 174 females, 75 males, and 36 turtles of unknown sex. Twenty-two females were gravid.

Blood samples were collected from the subcarapacial sinus due to its relative ease of access and minimal restraint requirement in a field setting. Grossly hemolyzed or lymph contaminated samples were removed from the study, although microscopic hemolysis cannot be ruled out. The timing of sampling relative to food consumption is unknown, so post-prandial changes, such as lipemia, could impact results. Physical examination was largely unremarkable for most individuals. Clinical signs of upper respiratory disease (URD), including oculonasal discharge, blepharoedema, and/or oral plaques were present in eleven turtles. Integumentary abnormalities including abrasions, lacerations, and/or nodules were present in 28 turtles. Appendicular abnormalities including abnormal nails and/or missing digits, feet, limbs, or tail tips were present in 42 animals. Cloacal abnormalities consisting of erythema, swelling, and/or discharge were present in 15 animals. Shell abnormalities involved the carapace (erosions– 25, predator injury—8) and plastron (erosions– 99, predator injury—12). In total, 30 turtles had active physical exam abnormalities significant enough to compromise health, and these individuals were excluded from the reference interval dataset.

All absolute EPH parameters varied by year, with TP (effect size = 3.70g/L, 95% CI = 1.60–5.90g/L, p = 0.01), albumin (effect size = 1.20g/L, 95% CI = 0.70–1.70g/L, p < 0.01), alpha 1 globulins (effect size = 0.86g/L, 95% CI = 0.60–1.10g/L, p < 0.01) alpha 2 globulins (effect size = 0.63g/L, 95% CI = 0.26–1.00g/L, p = 0.01), and gamma globulins (effect size = 4.30g/L, 95% CI = 3.70–4.80g/L, p < 0.01) higher in 2018 than 2019, and A:G (effect size = 0.02, 95% CI = 0.01–0.03, p = 0.02), prealbumin (effect size = 0.67g/L, 95% CI = 0.58–0.77g/L, p < 0.01), and beta globulins (effect size = 2.60g/L, 95% CI = 1.60–3.60g/L, p = 0.01) higher in 2019 than 2018. The relationships between relative EPH fraction and year were similar, except there was no significant association between relative alpha 2 globulins and year (p = 0.34).

The effects of location on absolute EPH parameters depended on year (significant Year*Location interaction, p < 0.05) for TP, A:G, prealbumin, albumin, and gamma globulins (Fig 2). A significant year*location effect was also identified for relative prealbumin (Fig 3). Location influenced absolute alpha 2 globulins (p = 0.02), absolute beta globulins (p = 0.04), and relative gamma globulins (p = 0.01) independent of year, while it was not a statistically significant predictor of absolute alpha 1 globulins, relative albumin, relative alpha 1 globulins, relative alpha 2 globulins, or relative beta globulins (p > 0.05). In addition to the site-level effects identified using general linear models, finer-scale clusters of high and low absolute EPH values were identified using spatial modeling (Fig 4). The location of these spatial clusters varied

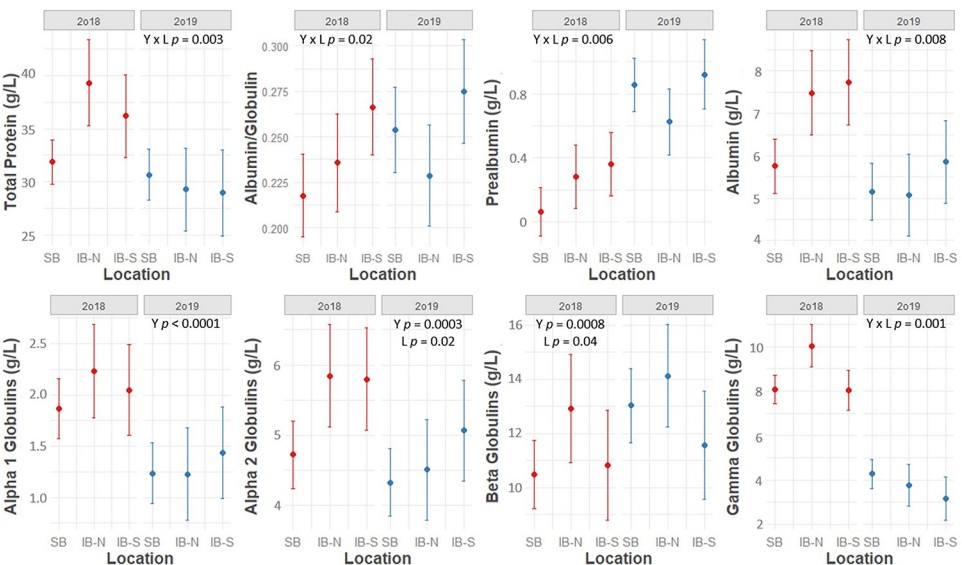

**Fig 2. Model predictions for plasma protein electrophoresis values based on year and location.** Model predictions with 95% confidence intervals for plasma protein electrophoresis values in free-living Blanding's turtles (*Emydoidea blandingii*) based on year and location. Model estimates were produced by top-ranking general linear mixed models (see Table 2). SB = Spring Bluff-Chiwaukee Prairie, IB-N = Illinois Beach North, IB-S = Illinois Beach South.

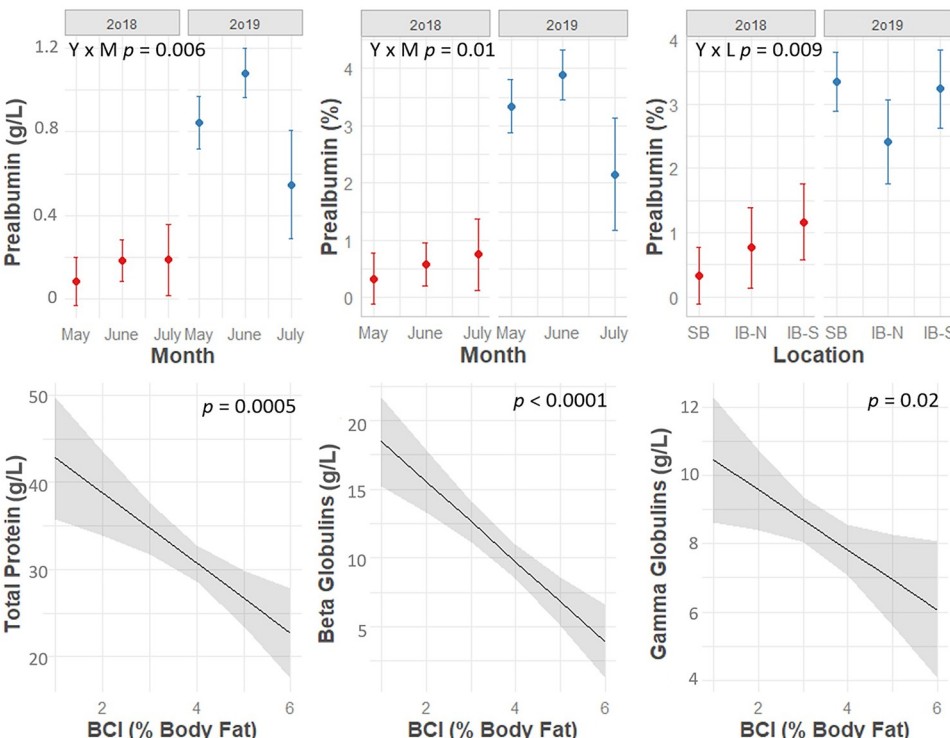

**Fig 3. Model predictions for plasma protein electrophoresis values based on year, location, month, and BCI.** Model predictions with 95% confidence intervals for plasma protein electrophoresis values in free-living Blanding's turtles (*Emydoidea blandingii*) based on year, location, month, and body condition index (BCI). Model estimates were produced by top-ranking general linear mixed models (see Table 2).

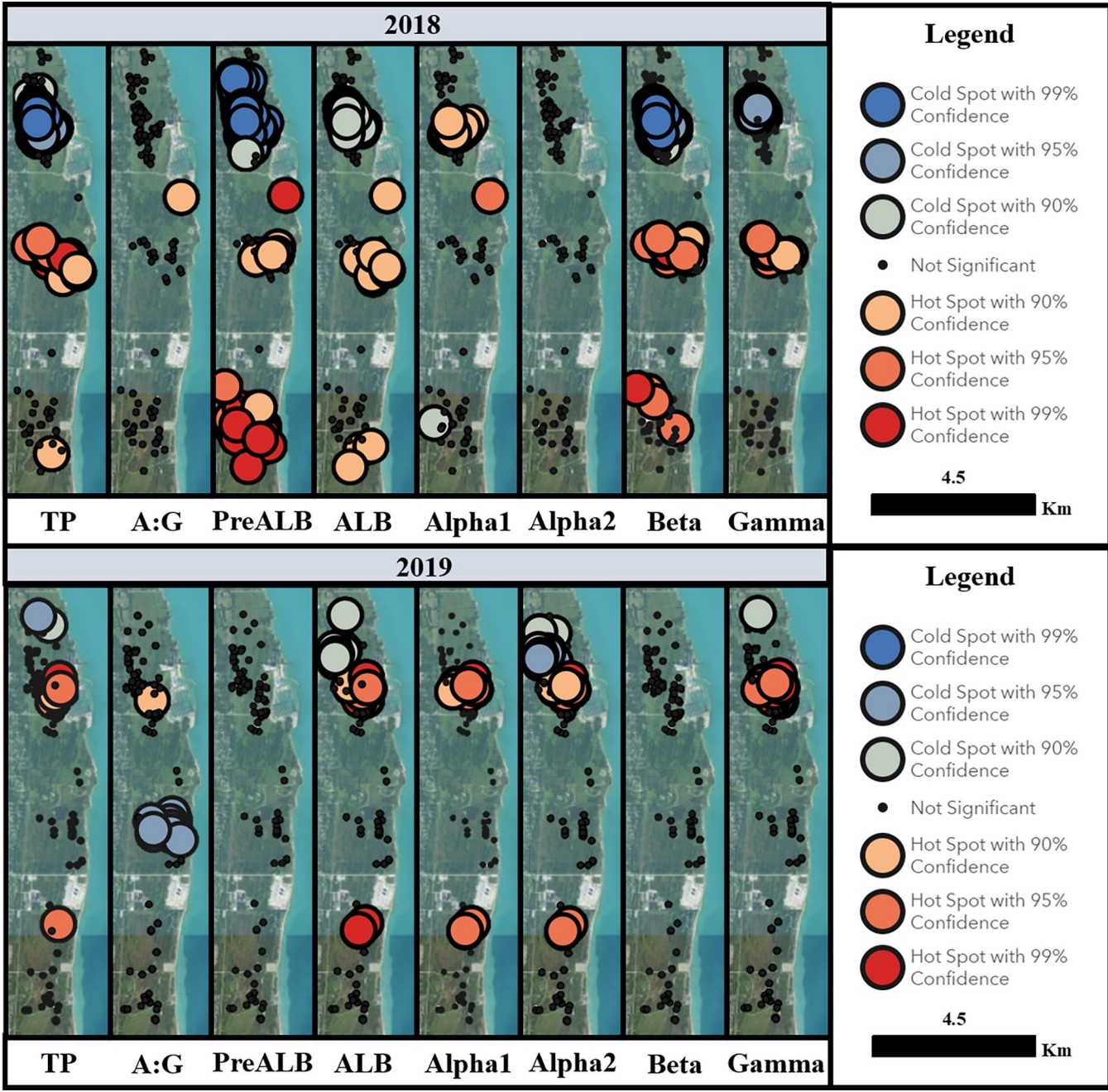

**Fig 4. 2018 and 2019 spatial clusters of plasma protein electrophoresis values in free-living Blanding's turtles (*Emydoidea blandingii*).** Study site map displaying hot and cold spots for each analyte in 2018 and 2019. Hot spots are areas with higher plasma protein concentrations, while cold spots are areas with lower plasma protein concentrations. SBCP, the managed site, is the northernmost territory on the map followed by the North unit of IBSP as the central territory and the South unit of IBSP as the southernmost territory. Evaluated using Gettis-Ord Gi* models. Map obtained from USGS National Map Viewer. TP = total protein (g/L), PreALB = prealbumin (g/L), ALB = albumin (g/L), Alpha1 = alpha 1 globulins (g/L), Alpha2 = alpha 2 globulins (g/L), Beta = beta globulins (g/L), Gamma = gamma globulins (g/L), A:G = albumin / globulin.

between years, and differences occurred both between and within study sites, especially in turtles sampled at SBCP in 2019 (Fig 4).

A:G ($p < 0.01$), absolute albumin ($p = 0.01$), relative albumin ($p < 0.01$), and relative gamma globulins ($p = 0.02$) differed by month, while the effects of month on absolute and

**Table 1. Protein electrophoresis values that vary by month and the presence of plastron abnormalities in free-living Blanding's turtles (*Emydoidea blandingii*).** Model estimates were produced by top-ranking general linear mixed models (see Table 2).

| Analyte | Level | Model Estimate | SE | Contrast | Difference | 95% CI | P—value |
|---|---|---|---|---|---|---|---|
| **Month** | | | | | | | |
| **Albumin/Globulin** | May | 0.27 | 0.01 | May vs. June | -0.04 | -0.06, -0.02 | < 0.01 |
| | June | 0.31 | 0.01 | June vs. July | 0.04 | 0.01, 0.06 | 0.01 |
| | July | 0.27 | 0.01 | May vs. July | -0.01 | -0.03, 0.02 | 1.00 |
| **Albumin (g/L)** | May | 6.00 | 0.40 | May vs. June | -1.40 | -2.20, -0.60 | 0.01 |
| | June | 7.40 | 0.40 | June vs. July | 1.33 | 0.20, 2.30 | 0.01 |
| | July | 6.10 | 0.50 | May vs. July | -0.10 | -1.10, 0.90 | 1.00 |
| **Albumin (%)** | May | 19.00 | 0.64 | May vs. June | -2.10 | -3.30, -0.95 | 0.01 |
| | June | 21.10 | 0.44 | June vs. July | 1.80 | 0.33, 3.20 | 0.01 |
| | July | 19.40 | 0.68 | May vs. July | -0.36 | -1.70, 0.98 | 0.80 |
| **Gamma Globulins (%)** | May | 17.20 | 0.41 | May vs. June | 1.31 | 0.20, 2.40 | 0.02 |
| | June | 15.90 | 0.39 | June vs. July | -0.81 | -2.30, 0.70 | 0.40 |
| | July | 16.70 | 0.58 | May vs. July | 0.51 | -0.97, 2.00 | 0.70 |
| **Plastron** | | | | | | | |
| **Beta Globulins (g/L)** | Normal | 11.30 | 0.70 | N vs. E | -1.60 | -2.90, -0.40 | 0.01 |
| | Erosions | 12.90 | 0.70 | E vs. I | -2.10 | -4.90, 0.70 | 0.15 |
| | Injury | 15.00 | 1.40 | N vs. I | -3.70 | -6.60, -0.90 | 0.01 |
| **Beta Globulins (%)** | Normal | 35.60 | 0.80 | N vs. E | -0.27 | -1.64, 1.10 | 0.70 |
| | Erosions | 35.90 | 0.90 | E vs. I | -3.33 | -6.40, -0.28 | 0.03 |
| | Injury | 39.20 | 1.50 | N vs. I | -3.60 | -6.60, -0.61 | 0.02 |

SE = standard error. N = normal. E = erosions. I = injury.

relative prealbumin concentration depended on year (significant Month*Year interaction) (Fig 2, Tables 1 and 2). Specifically, relative gamma globulins were significantly higher in May compared to June while A:G, absolute albumin, and relative albumin were significantly higher

**Table 2. Top models for predicting protein electrophoresis values in free-living Blanding's turtles (*Emydoidea blandingii*) based on Akaike's information criterion, corrected for sample size (AIC$_c$).**

| Analyte | Model | N | K | AIC$_c$ | w$_i$ |
|---|---|---|---|---|---|
| **Total Protein (g/L)** | Y * L + BCI + Integ + App | 265 | 11 | 748.00 | 0.98 |
| **Albumin / Globulin** | Y * L + M + Sex + Age + Cloaca | 279 | 15 | -825.40 | 0.98 |
| **Prealbumin (g/L)** | Y * L + Y * M + URD | 279 | 13 | -995.90 | 0.80 |
| **Prealbumin (%)** | Y * L + Y * M + URD | 279 | 13 | -1534.60 | 0.69 |
| **Albumin (g/L)** | Y * L + M + App + Gravid | 278 | 12 | -7.70 | 0.57 |
| **Albumin (%)** | Y + M + Age + Sex + Gravid | 279 | 11 | -1154.10 | 0.94 |
| **Alpha 1 Globulins (g/L)** | Y + Age + Carapace | 279 | 10 | -423.10 | 0.58 |
| **Alpha 1 Globulins (%)** | Y + Age + Carapace | 279 | 10 | -1232.30 | 0.80 |
| **Alpha 2 Globulins (g/L)** | Y + L + Age | 283 | 8 | -157.30 | 0.76 |
| **Alpha 2 Globulins (%)** | Sex + Age + Gravid | 280 | 8 | -1207.10 | 0.95 |
| **Beta Globulins (g/L)** | Y + L + Sex + BCI + Integ + Plastron + App | 265 | 15 | 307.80 | 0.99 |
| **Beta Globulins (%)** | Y + Age + Sex + Plastron + Gravid | 280 | 13 | -891.00 | 0.99 |
| **Gamma Globulins (g/L)** | Y * L + Age + BCI + App | 266 | 12 | -34.80 | 0.99 |
| **Gamma Globulins (%)** | Y + L + M + Age | 283 | 10 | -1129.80 | 0.86 |

Y = year, L = location, M = month, BCI = body condition index, Integ = integument, App = appendages, URD = upper respiratory disease.

**Table 3. Protein electrophoresis values that vary by age class in free-living Blanding's turtles (*Emydoidea blandingii*).** Model estimates were produced by top-ranking general linear mixed models (see Table 2).

| Analyte | Level | Model Estimate | SE | Contrast | Difference | 95% CI | P—value |
|---|---|---|---|---|---|---|---|
| **Albumin/Globulin** | Adult | 0.25 | 0.01 | Ad vs. Juv | -0.07 | -0.11, -0.04 | < 0.01 |
| | Sub-Adult | 0.28 | 0.01 | Ad vs. SA | -0.04 | -0.06, -0.01 | 0.01 |
| | Juvenile | 0.32 | 0.02 | SA vs. J | -0.04 | -0.07, -0.01 | 0.03 |
| **Albumin (%)** | Adult | 17.90 | 0.41 | Ad vs. Juv | -3.80 | -5.80, -1.80 | <0.01 |
| | Sub-Adult | 19.80 | 0.63 | Ad vs. SA | -1.80 | -3.10, -0.56 | 0.01 |
| | Juvenile | 21.80 | 0.87 | SA vs. J | -2.00 | -4.00, 0.02 | 0.05 |
| **Alpha 1 Globulins (g/L)** | Adult | 1.60 | 0.10 | Ad vs. Juv | -2.40 | -2.90, -1.80 | < 0.01 |
| | Sub-Adult | 2.70 | 0.10 | Ad vs. SA | -1.00 | -1.40, -0.70 | < 0.01 |
| | Juvenile | 4.00 | 0.20 | SA vs. J | -1.30 | -1.90, -0.80 | < 0.01 |
| **Alpha 1 Globulins (%)** | Adult | 5.20 | 0.30 | Ad vs. Juv | -9.30 | -10.60, -8.00 | < 0.01 |
| | Sub-Adult | 8.70 | 0.40 | Ad vs. SA | -3.50 | -4.40, -2.60 | < 0.01 |
| | Juvenile | 14.50 | 0.60 | SA vs. J | -5.80 | -7.20, -4.40 | < 0.01 |
| **Alpha 2 Globulins (g/L)** | Adult | 5.10 | 0.20 | Ad vs. Juv | 0.10 | -0.70, 0.90 | 1.00 |
| | Sub-Adult | 5.80 | 0.30 | Ad vs. SA | -0.70 | -1.30, -0.20 | 0.01 |
| | Juvenile | 5.00 | 0.40 | SA vs. J | 0.80 | 0.04, 1.70 | 0.04 |
| **Alpha 2 Globulins (%)** | Adult | 14.30 | 0.37 | Ad vs. Juv | -1.10 | -2.90, 0.60 | 0.30 |
| | Sub-Adult | 16.40 | 0.50 | Ad vs. SA | -2.10 | -3.20, -1.00 | < 0.01 |
| | Juvenile | 15.50 | 0.78 | SA vs. J | 1.00 | -0.90, 2.80 | 0.40 |
| **Beta Globulins (%)** | Adult | 41.70 | 0.79 | Ad vs. Juv | 9.61 | 6.40, -12.83 | < 0.01 |
| | Sub-Adult | 34.40 | 0.97 | Ad vs. SA | 7.28 | 5.40, -9.16 | < 0.01 |
| | Juvenile | 32.10 | 1.50 | SA vs. J | 2.33 | -0.98, 5.65 | 0.20 |
| **Gamma Globulins (g/L)** | Adult | 7.00 | 0.30 | Ad vs. Juv | 2.00 | 0.60, 3.40 | 0.01 |
| | Sub-Adult | 6.60 | 0.40 | Ad vs. SA | 0.30 | -0.80, 1.40 | 0.85 |
| | Juvenile | 5.00 | 0.50 | SA vs. J | 1.70 | 0.60–3.00 | 0.01 |
| **Gamma Globulins (%)** | Adult | 19.00 | 0.31 | Ad vs. Juv | 6.00 | 4.37, 7.63 | < 0.01 |
| | Sub-Adult | 18.00 | 0.45 | Ad vs. SA | 0.98 | -0.22, 2.20 | 0.13 |
| | Juvenile | 13.00 | 0.65 | SA vs. J | 5.00 | 3.36, 6.68 | < 0.01 |

SE = standard error. Ad = adult. SA = sub-adult. Juv = juvenile.

in June compared to both May and July. Relative and absolute prealbumin followed a similar trend to A:G and albumin in 2019, but values were not significantly different by month in 2018.

Age class influenced A:G and all relative EPH fractions except prealbumin; it was also found to be an important predictor of absolute alpha 1 globulins, alpha 2 globulins, and gamma globulins (Table 3). Juvenile turtles had the highest values for A:G, relative albumin, relative alpha 1 globulins, and absolute alpha 1 globulins. Subadults had the highest values for relative and absolute alpha 2 globulins. Adults had the highest values for relative beta globulins, relative gamma globulins, and absolute gamma globulins.

Sex influenced the A:G, relative albumin, relative alpha 2 globulins, relative beta globulins, and absolute beta globulins (Table 4). Specifically, male turtles had higher A:G, relative albumin, and relative alpha 2 globulins, while females had higher absolute and relative beta globulin values. Gravid females had higher relative beta globulins and lower A:G, absolute and relative albumin, and alpha 2 globulins.

Multiple EPH fractions were associated with health predictors. Body condition was negatively associated with TP, absolute beta globulins, and absolute gamma globulins (Fig 2).

**Table 4. Protein electrophoresis values that vary by sex and gravidity in free-living Blanding's turtles (*Emydoidea blandingii*).** Model estimates were produced by top-ranking general linear mixed models (see Table 2).

| Predictor | Analyte | Level | Model Estimate | SE | Difference | 95% CI | P—value |
|---|---|---|---|---|---|---|---|
| **Sex** | **Albumin (%)** | Male | 20.40 | 0.60 | 1.21 | 0.31–2.11 | 0.01 |
| | | Female | 19.20 | 0.50 | | | |
| | **Albumin/Globulin** | Male | 0.30 | 0.01 | 0.02 | 0.01–0.04 | 0.01 |
| | | Female | 0.27 | 0.01 | | | |
| | **Alpha 2 Globulins (%)** | Male | 16.20 | 0.50 | 1.60 | 0.72–2.40 | 0.01 |
| | | Female | 14.60 | 0.43 | | | |
| | **Beta Globulins (g/L)** | Male | 12.30 | 0.90 | 1.80 | 0.50–3.20 | 0.01 |
| | | Female | 14.10 | 0.80 | | | |
| | **Beta Globulins (%)** | Male | 34.50 | 0.97 | 3.04 | 1.56–4.53 | < 0.01 |
| | | Female | 37.60 | 0.92 | | | |
| **Gravidity** | **Albumin/Globulin**[a] | Non-Gravid | 0.31 | 0.01 | 0.03 | 0.02–0.06 | 0.04 |
| | | Gravid | 0.27 | 0.02 | | | |
| | **Albumin (g/L)** | Non-Gravid | 7.10 | 0.30 | 1.20 | 0.03–2.30 | 0.04 |
| | | Gravid | 6.00 | 0.60 | | | |
| | **Albumin (%)** | Non-Gravid | 21.00 | 0.32 | 2.42 | 0.88–3.95 | 0.01 |
| | | Gravid | 18.60 | 0.84 | | | |
| | **Alpha 2 Globulins (%)** | Non-Gravid | 16.40 | 0.28 | 1.90 | 0.63–3.24 | 0.01 |
| | | Gravid | 14.40 | 0.70 | | | |
| | **Beta Globulins (%)** | Non-Gravid | 34.40 | 0.73 | 3.40 | 1.07–5.71 | 0.01 |
| | | Gravid | 37.80 | 1.30 | | | |

SE = standard error.

[a] Gravidity was not included in the top-ranking model for Albumin/Globulin due to confounding with the "cloaca" variable, but considered separately it is significantly associated with Albumin/Globulin.

Turtles with predator injuries or erosions of the plastron had higher relative and absolute beta globulins than those with normal plastrons (Table 1). Abnormalities of the appendicular system were associated with higher TP and absolute albumin, beta globulins, and gamma globulins (Table 5). Integumentary abnormalities were associated with lower TP and absolute beta globulins (Table 5). Turtles with upper respiratory disease had lower absolute and relative prealbumin than those without (Table 5). Finally, cloacal abnormalities were associated with a lower A:G (Table 5). Top models for each EPH fraction tended to include spatiotemporal, demographic, and health predictors, highlighting the influence that each of these components has on the distribution of blood proteins in Blanding's turtles (Table 2).

Reference intervals were constructed in both a subject-based and population-based manner for each EPH analyte (Table 6). Based on the index of individuality, subject-based reference intervals are superior for absolute and relative alpha 1 globulins, absolute beta globulins, and relative albumin in Blanding's turtles. The remainder of the II values fell between 0.6 and 1.4, indicating that population-based reference intervals were employed and interpreted with caution for all other fractions.

## Discussion

We set out to describe baseline plasma protein fractions in a well-studied population of Blanding's turtles in northeastern Illinois and southeastern Wisconsin and observed EPH fractions varied significantly based on spatiotemporal, demographic, and health factors. Understanding

**Table 5. Protein electrophoresis values that vary based on the presence of physical examination abnormalities in free-living Blanding's turtles (*Emydoidea blandingii*).** Model estimates were produced by top-ranking general linear mixed models (see Table 2). Top models for relative and absolute alpha 1 globulins contained "carapace", however no contrasts for this predictor were significant therefore not reported.

| Predictor | Analyte | Level | Model Estimate | SE | Difference | 95% CI | P—value |
|---|---|---|---|---|---|---|---|
| **Appendages** | **Total Protein (g/L)** | Normal | 31.90 | 1.10 | 5.30 | 1.90–8.06 | 0.01 |
| | | Abnormal | 37.10 | 1.80 | | | |
| | **Albumin (g/L)** | Normal | 6.10 | 0.30 | 9.00 | 0.10–1.70 | 0.03 |
| | | Abnormal | 7.00 | 0.50 | | | |
| | **Beta Globulins (g/L)** | Normal | 12.10 | 0.70 | 2.20 | 0.50–3.80 | 0.01 |
| | | Abnormal | 14.20 | 1.00 | | | |
| | **Gamma Globulins (g/L)** | Normal | 5.50 | 0.30 | 1.40 | 0.60–2.10 | 0.01 |
| | | Abnormal | 6.80 | 0.40 | | | |
| **Integument** | **Total Protein (g/L)** | Normal | 31.90 | 1.10 | 4.60 | 0.70–8.50 | 0.02 |
| | | Abnormal | 27.20 | 2.00 | | | |
| | **Beta Globulins (g/L)** | Normal | 14.10 | 0.70 | 2.00 | 0.20–3.80 | 0.03 |
| | | Abnormal | 12.20 | 1.10 | | | |
| **Upper Respiratory Disease** | **Prealbumin (g/L)** | Absent | 0.58 | 0.03 | 0.27 | 0.03–0.52 | 0.03 |
| | | Present | 0.31 | 0.12 | | | |
| | **Prealbumin (%)** | Absent | 1.88 | 0.13 | 1.04 | 0.11–1.97 | 0.03 |
| | | Present | 0.84 | 0.48 | | | |
| **Cloaca** | **Albumin/Globulin** | Normal | 0.31 | 0.01 | 0.04 | 0.01–0.07 | 0.01 |
| | | Abnormal | 0.26 | 0.02 | | | |

SE = standard error.

how reptile clinical pathology values correlate to landscape changes is important for contextualizing health assessments and evaluating ecosystem wellness [1, 2, 5, 6].

We documented several statistically significant inter-annual differences in Blanding's turtle total protein and EPH fractions. These changes may be attributable to fluctuation in climactic variables that influence turtle metabolism and resource availability. Temperature is a key determinant of metabolic rates in ectotherms, including reptiles [41, 42], and previous studies in loggerheads (*Caretta caretta*) and green turtles (*Chelonia mydas*) observed a negative correlation with the A:G and environmental temperature [16]. In Lake County, the average air temperatures in May (60.4˚F/15.8˚C) and June (66.4˚F/19.1˚C) of 2018 were warmer than those in May (54.7˚F/12.6˚C) and June (64.2˚F/17.9˚C) of 2019 [43]. Similar to the temperature-associated protein changes in sea turtles, Blanding's turtle TP, albumin, alpha 1 and alpha 2 globulins, and gamma globulin concentrations were greater in 2018, while prealbumin, beta globulin, and A:G were greater in 2019. It is plausible that turtles would be more active, consume more food items, initiate reproductive activity, and mount immune responses more efficiently at higher temperatures, all of which may increase circulating protein concentrations and contribute to the observed inter-annual variability in EPH fractions [41].

Temperature, however, is not the only environmental factor that differed between years. Rainfall was also greater during May and June of 2018 (30.2 inches total) compared to the same time period in 2019 (18.76 inches total), while humidity was a bit more consistent between years (77.2% average relative humidity in May and June of 2018 and 80.75% average relative humidity in May and June of 2019) [43]. Water availability and humidity both influence behavioral thermoregulation in ectotherms and can modify their activity levels in complex ways [44]. Increased activity secondary to rainfall may create more opportunities for antigenic stimulation and contribute to changes in food consumption, reproductive behaviors,

**Table 6. Summary data including data distribution, measure of central tendency (mean for normally distributed variables, median for non-normally distributed variables), measure of dispersion (standard deviation for normally distributed variables, 10th–90th percentiles for non-normally distributed variables), and reference intervals for plasma protein electrophoresis in free-living, apparently healthy Blanding's turtles (*Emydoidea blandingii*).**

| Analyte | N | Dist | CT | Disp | Min | Max | Reference Interval | 90% CI LB | 90% CI UB | II | RCV (%) | Sub vs. Pop |
|---|---|---|---|---|---|---|---|---|---|---|---|---|
| **Total Protein (g/L)** | 193[a] | NG | 31.00 | 18.00–47.50 | 13.00 | 56.00 | 15.70–52.30 | 15.40–17.60 | 48.60–53.60 | 0.68 | 62.70 | Pop |
| **Albumin/Globulin** | 195[b] | G | 0.29 | 0.07 | 0.14 | 0.46 | 0.17–0.43 | 0.16–0.19 | 0.40–0.46 | 0.66 | 41.20 | Pop |
| **Prealbumin (g/L)** | 196 | NG | 0.40 | 0.05–1.20 | 0.00 | 2.30 | 0.00–2.00 | 0.00–0.10 | 1.80–2.30 | 0.90 | 237.00 | Pop |
| **Albumin (g/L)** | 196 | NG | 6.30 | 3.40–9.90 | 1.90 | 15.60 | 2.50–13.40 | 2.10–3.00 | 12.90–15.00 | 0.66 | 74.40 | Pop |
| **Alpha 1 Globulins (g/L)** | 196 | NG | 1.90 | 1.00–4.30 | 0.70 | 11.60 | 0.80–5.70 | 0.70–0.80 | 4.10–6.10 | 0.54 | 88.40 | Sub |
| **Alpha 2 Globulins (g/L)** | 196 | NG | 4.70 | 2.70–7.40 | 1.70 | 11.20 | 2.10–10.00 | 1.90–2.50 | 9.60–10.70 | 0.67 | 70.80 | Pop |
| **Beta Globulins (g/L)** | 196 | NG | 10.80 | 5.04–17.90 | 3.40 | 30.00 | 4.00–25.70 | 3.30–4.40 | 21.80–29.90 | 0.56 | 66.70 | Sub |
| **Gamma Globulins (g/L)** | 195[c] | NG | 5.60 | 2.80–10.70 | 1.40 | 16.50 | 1.70–14.10 | 1.30–1.90 | 12.30–15.70 | 0.99 | 144.00 | Pop |
| **Prealbumin (%)** | 196 | NG | 1.00 | 0.00–4.20 | 0.00 | 10.10 | 0.00–6.70 | 0.00–1.00 | 4.90–7.80 | 0.93 | 266.00 | Pop |
| **Albumin (%)** | 195[d] | G | 20.30 | 3.80 | 13.00 | 31.00 | 14.00–29.00 | 13.00–15.00 | 27.00–31.00 | 0.59 | 29.90 | Sub |
| **Alpha 1 Globulins (%)** | 196 | NG | 6.00 | 4.00–14.00 | 2.70 | 28.00 | 3.00–23.00 | 2.90–3.30 | 19.00–28.90 | 0.34 | 51.80 | Sub |
| **Alpha 2 Globulins (%)** | 195[e] | NG | 16.00 | 11.00–19.70 | 9.00 | 24.00 | 10.00–21.70 | 9.70–11.00 | 20.50–23.30 | 0.62 | 31.20 | Pop |
| **Beta Globulins (%)** | 196 | NG | 35.80 | 25.00–49.10 | 18.00 | 59.10 | 20.00–55.20 | 19.00–21.00 | 51.90–57.70 | 0.82 | 59.20 | Pop |
| **Gamma Globulins (%)** | 196 | NG | 19.00 | 11.30–28.00 | 8.00 | 37.00 | 9.700–32.10 | 9.40–10.00 | 31.20–34.10 | 1.20 | 114.00 | Pop |

Dist = distribution, NG = Non-Gaussian, G = Gaussian, CT = measure of central tendency, Disp = measure of dispersion, CI = confidence interval, LB = lower bound of reference interval, UB = upper bound of reference interval, II = index of individuality, RCV = reference change value, Sub = subject-based refence interval recommended, Pop = population-based reference interval recommended.

[a] Outliers removed: 10.00, 71.00, 74.00 g/L.

[b] Outliers removed: 0.51.

[c] Outliers removed: 22.6 g/L.

[d] Outliers removed: 11%.

[e] Outliers removed: 26%.

and other physiologic processes with a resultant increase in plasma protein concentrations [44]. While the effects of some climactic variables on ectotherm physiology have been at least partially characterized, many other environmental variables may also impact resource availability and overall wellness. Additional research is needed to determine the underlying environmental causes of temporal variability in reptile protein electrophoresis values. It is likely that inter-annual differences in protein electrophoresis values also exist for other reptiles [45]. Unfortunately, direct comparison to existing literature is difficult, because although some multi-year chelonian EPH studies exist [4, 10, 16, 18, 46–52], the possibility of inter-annual variation in EPH values is infrequently assessed. Our findings indicate that future EPH studies in reptiles should consider the potential for significant inter-annual effects.

TP and absolute EPH fractions differed by location in Blanding's turtles, similar to eastern box turtles (*Terrapene carolina carolina*) [4], alligator snapping turtles (*Macrochelys temminckii*) [46] and green turtles [45]. SBCP and IBSP vary significantly in their habitats despite their close proximity. SBCP offers a coastal dune and swale habitat that has been managed and preserved since 2004, whereas IBSP includes adjacent north (N) and south (S) units containing dune, prairie, oak-savannah and wetland habitats, with limited management prior to 2017 [22]. The effect of location on EPH parameters depended on the year; and the differences in EPH values between the study sites and years were rarely consistent. The only consistent location-related differences between 2018 and 2019 were the cold spots at the northern end of SBCP, potentially indicating a more consistent health status for turtles occupying this area. Inter-annual fluctuation in TP and EPH fractions between and within the different study sites

may indicate transient, localized changes in health status with unclear management implications. Focused, longitudinal assessment of telemetered individuals within these areas will be useful to identify biotic and abiotic factors associated with acute changes in EPH parameters. Our findings indicate that it is important to consider both location and time at multiple different scales in order to obtain a nuanced understanding of the drivers of turtle health status.

Several Blanding's turtle plasma protein values were affected by month; specifically, A:G, absolute albumin, relative albumin, and both absolute and relative prealbumin values peaked in June, while relative gamma globulins were highest in May. Similar patterns have been documented in other chelonians including Hermann's tortoises (*Testudo hermanni*) [53], alligator snapping turtles [46], and eastern box turtles [4]. These changes may be attributed to increased interactions with other turtles during the mating season and/or reproductive physiology. In the Lake Plain, Blanding's turtle mating season occurs from March to May, while nesting season typically begins in June. Turtles are more likely to interact with each other and potentially transmit pathogens such as *Emydoidea* herpesvirus 1 during mating season [54], which may contribute to an increase in gamma globulins (i.e. immunoglobulins). Alternatively, the increased gamma globulin concentration in May could be secondary to vitellogenesis, as the release of estrogen stimulates hyperglobulinemia in chelonians [55, 56]. Elevated albumin concentrations in June may be attributable to dehydration associated with prolonged overland trips and nesting-associated exertion, similar to previous reports in sea turtle species [48, 57–59]. Studies in other chelonian and lizard species have documented elevations in albumin and total proteins during the summer months and have correlated these elevations to the increased food consumption and reproductive activity in this time period [4, 60, 61].

Gravid females had higher relative beta globulins and lower A:G, absolute and relative albumin, and alpha 2 globulins. As reviewed above, gravid reptiles can develop hyperglobulinemia during vitellogenesis in response to estrogen [62]. A study conducted in pond sliders (*Trachemys scripta*) demonstrated that estrogen also downregulates albumin, which may be a factor in the lower albumin concentrations observed in gravid female Blanding's turtles [63]. Higher globulins and lower albumin secondary to estrogen production would also support the lower A:G in gravid females. In birds, elevated beta globulins are attributed to egg production [64]. In leatherbacks, alpha 2 globulins decrease over the nesting season due to inanition [65]. Our findings are important to provide context for future studies on EPH in gravid chelonians, as several of the changes associated with gravidity (elevated beta globulins, lower albumin and A:G) can also be interpreted as indicative of inflammation and poor health [17].

Male turtles had higher A:G, relative albumin, and relative alpha 2 globulins while females had higher absolute and relative beta globulin values. Many of these findings differ from those in other chelonians. Relative albumin was higher in female red-eared sliders (*Trachemys scripta elegans*) and map turtles (*Graptemys geographica*) [66], and absolute albumin was higher in female eastern box turtles [4]. Male loggerhead sea turtles had higher absolute and relative beta globulin concentrations, although there was a great deal of beta-gamma bridging indicating possible underlying disease processes in those individuals [10]. Male radiated tortoises had higher relative alpha 2 globulins during winter sampling, and female eastern box turtles and radiated tortoises had higher absolute and relative beta globulin concentrations [4, 67]. Consistent with a previous study in this population of Blanding's, but contrary to several other studies in chelonians, there was no difference in total protein between the sexes [24]. Many of these findings may be confounded by the timing of our sampling, since it was concentrated during the breeding and nesting season. It is possible that if these turtles were sampled later in the year we would find that sex-based differences in Blanding's turtles are more in line with what is reported for other species. Blanding's turtles have some unique sex-associated

EPH patterns compared to other chelonians, underscoring the need for studies like this to understand species differences in clinical pathology values.

Protein fraction concentrations varied across age groups, with juvenile turtles having the highest values for A:G, relative albumin, relative alpha 1 globulins, and absolute alpha 1 globulins. Similar trends are seen in juvenile loggerheads [16], juvenile Kemp's ridley sea turtles (*Lepidochelys kempii*) [68, 69], and juvenile gopher tortoises (*Gopherus polyphemus*) [70]. Adults had the highest values for relative beta globulins, relative gamma globulins, and absolute gamma globulins, which are similar to findings in other chelonians. Adult loggerheads [10], eastern box [4], and green turtles [49] all had higher absolute beta globulin levels, and adult Kemp's ridley sea turtles [69] had higher absolute beta and gamma globulins compared to their juvenile counterparts. In general, the changes found can be attributed to increased antigenic challenge as turtles age and become reproductively mature, a consistent finding in other chelonian studies [16, 19, 46].

The overall health status of individuals also contributed to variations in plasma protein concentrations. Higher absolute beta and gamma globulins were found in turtles with lower body fat percentage, which both increase in the presence of acute and chronic inflammation. BCS is a reliable measure of health status in other reptiles, with a good body condition score equating to better immune function and capability to fight periodic bouts of disease [71]. The elevated relative and absolute beta globulins in turtles with plastron injuries and erosions might be due to the plastron constantly being in contact with either unclean water or the ground, increasing the potential for chronic antigenic stimulation when injuries or abnormalities are present.

Appendage abnormalities were associated with higher TP and absolute beta and gamma globulins, indicating the presence of possible chronic inflammation. The loss of nails or digits from a variety of causes results in open wounds where infections may develop. Missing nails and digits might also have an impact on the turtle's overall health, making tasks like swimming and foraging more difficult [72]. In eastern box turtles, microvascular problems and primary microbial infections can cause the loss of digits and nails, indicating that even apparently minor anatomical abnormalities may have physiologically significant impacts on these turtles [73]. Integument abnormalities were associated with lower TP and absolute beta globulin concentrations. A study conducted in green and loggerhead sea turtles with traumatic wounds to their carapace, head, and/or flippers showed similar trends, with those experiencing trauma having lower beta globulin concentrations than their healthy counterparts [16].

Turtles with evidence of upper respiratory disease had lower relative prealbumin concentrations. Like albumin, prealbumin is a negative acute phase protein and decreases in the presence of inflammation [74]. Prealbumin concentrations can also be lower in cases of protein malnutrition [74]. These turtles may have lower concentrations because they are ill with upper respiratory infection, and this illness could be preventing them from taking in an adequate amount of protein. It is important to note, however, that prealbumin has not yet been validated in chelonian species, so it is unclear if prealbumin is truly what is represented in the prealbumin region of the electrophoretogram. The lower A:G in turtles with cloacal abnormalities likely consistent with inflammation associated with infection, inflammation, or stress, with a lower ratio usually indicating hyperglobulinemia [75].

The population of Blanding's turtles that was studied showed a high degree of within-individual variability in EPH parameters at different points in time. This was also identified for hematologic and plasma biochemical parameters in the same population [24]. Reptile clinical pathology parameters are widely variable, and it is important to understand the many factors come together to influence the absolute value of each analyte. Our findings in Blanding's turtles indicate that the index of individuality and subsequent need for subject-based reference

intervals should be investigated in other reptile species in order to improve the interpretation of clinical pathology testing.

There are a few limitations in this study that could be addressed in future research. Due to radiotelemetry strategy in the Lake County location, there was a bias towards adult female turtles, which are followed closely to identify nest location. Additionally, a limited number of overtly unhealthy turtles were identified, with only a few individuals having physical examination abnormalities. While this is recognized as a positive finding considering that it indicates the population is doing well, it does limit the ability to determine how EPH values change in states of poor health. Furthermore, turtles that could be identified more than once over the course of the sampling period were sampled every three to six weeks, which could indicate the fluctuation of an inflammatory response or antigenic stimulation over time.

All blood samples were obtained from the subcarapacial sinus. Blood and lymphatic vessels are very closely associated with one another at this site, making it possible that blood samples could become contaminated with lymphatic fluid during venipuncture [76]. While samples with obvious lymph contamination were discarded, undetectable lymph contamination could have negatively impacted results. Lymph contamination has been known to falsely decrease PCV and hemoglobin concentrations and may have similar affects on plasma protein concentrations [77]. For those turtles that were collect and sampled from traps, the stress of trapping could ultimately play a role in affecting plasma protein fractions [50], but the significance has not been studied in Blanding's turtles. Timing samples to be collected pre- or post-prandial to account for lipemia is difficult to control in wildlife research, but a study conducted in Kemp's ridley and green sea turtles revealed that feeding had very minimal effects on plasma biochemical values and are therefore unlikely to alter clinical interpretation [78].

There is a discrepancy between the use of TS verses total protein (TP). A refractometer is an efficient way to measure TS in a field research setting, but TS includes both plasma proteins as well as additional plasma solutes [79]. While there have been multiple studies conducted in other chelonian species that show a significant correlation between TS and TP [19, 80], a direct relationship in Blanding's turtles has not been previously identified. Following separation, the collected plasma was frozen until protein electrophoresis could be run. Studies conducted in other reptile species have identified EPH differences in fresh plasma compared to frozen/thawed samples, and some of those studies recommended using fresh samples for best results [13, 81]. In our circumstances, it would have been impractical and cost-prohibitive to ship over 200 fresh plasma samples for analysis separately, underscoring the need for us to freeze and batch-run our samples. Hemolysis also has the potential to affect EPH values, but no grossly hemolyzed samples were use in this study [82].

The baseline plasma protein reference intervals generated in this study will be useful in defining the health status of this population. There is some overlap in the top models for predicting relative and absolute EPH fractions; however there also instances where the values vary, with absolute values being high while relative values are low for the same variable. This variation demonstrates the importance of considering both relative and absolute fractions when interpreting EPH values because they might be driven by different processes. Results of this study validate that month, location, sex, age class, and health status should be considered when interpreting EPH fractions. Although EPH does not provide details on specific diseases or stressors, it is a helpful tool that can aid in identifying when intervention and treatment might be needed. With the increased use of protein electrophoresis to evaluate the health status of animals in the veterinary medical field, the application of this tool in conservation of wild populations is becoming more widely accepted and studied. With baseline concentrations established and evaluated for variation, future studies can aim to validate and expand upon the normal reference intervals in this species.

## Supporting information

**S1 File. Combined data.** All project data collected upon turtle capture, which includes identifying information, temporal data, capture method, weather data, measurements, physical exam findings, and release information.
(XLSX)

## Author Contributions

**Conceptualization:** Kirsten E. Andersson, Laura Adamovicz, Gary Glowacki, Matthew C. Allender.

**Data curation:** Kirsten E. Andersson, Laura Adamovicz, Lauren E. Mumm, Samantha E. Bradley, Gary Glowacki, Matthew C. Allender.

**Formal analysis:** Laura Adamovicz, Matthew C. Allender.

**Funding acquisition:** Gary Glowacki.

**Investigation:** Kirsten E. Andersson, Laura Adamovicz, Lauren E. Mumm, Samantha E. Bradley, John M. Winter, Carolyn Cray, Matthew C. Allender.

**Methodology:** Kirsten E. Andersson, Laura Adamovicz, Lauren E. Mumm, Carolyn Cray, Matthew C. Allender.

**Project administration:** Kirsten E. Andersson, Laura Adamovicz, Lauren E. Mumm, Samantha E. Bradley, Matthew C. Allender.

**Resources:** Laura Adamovicz, Carolyn Cray, Matthew C. Allender.

**Software:** Laura Adamovicz, Matthew C. Allender.

**Supervision:** Kirsten E. Andersson, Laura Adamovicz, Matthew C. Allender.

**Validation:** Laura Adamovicz, Matthew C. Allender.

**Visualization:** Laura Adamovicz, Matthew C. Allender.

**Writing – original draft:** Kirsten E. Andersson, Laura Adamovicz.

**Writing – review & editing:** Kirsten E. Andersson, Laura Adamovicz, Lauren E. Mumm, Samantha E. Bradley, Gary Glowacki, Carolyn Cray, Matthew C. Allender.

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
