## [Decision Letter · Decision Letter 0]

2 Jul 2021

PONE-D-21-17850

Plasma electrophoresis profiles of Blanding’s turtles (Emydoidea blandingii) and influences of month, age, sex, health status, and location

PLOS ONE

Dear Dr. Andersson,

Thank you for submitting your manuscript to PLOS ONE. Your work was assessed by 2 subject experts and the Academic Editor. All 3 feel that your manuscript has merit and presents useful baseline data. However, sfter careful consideration, we feel that it has merit but does not fully meet PLOS ONE’s publication criteria as it currently stands. Therefore, we invite you to submit a revised version of the manuscript that addresses the points raised during the review process.

Required Revisions

1. Many parts of the Methods are not clear; the authors refer to telemetry, head-starting, etc., but these aspects are not described. Please carefully consider and respond to the specifics outlined in the comments provided by reviewer #2 in the uploaded manuscript file.

2. A sample electropherogram image should be provided as a figure to illustrate the quality of the work, as well as show how interpretation was done.

Suggested Revisions:

Please carefully consider and respond to all of the comments provided by the reviewers.

We look forward to receiving your revised manuscript.

Kind regards,

Christopher M. Somers

Academic Editor

PLOS ONE

Journal Requirements:

2. We note that Figures 3 and 4 in your submission contain map images which may be copyrighted. All PLOS content is published under the Creative Commons Attribution License (CC BY 4.0), which means that the manuscript, images, and Supporting Information files will be freely available online, and any third party is permitted to access, download, copy, distribute, and use these materials in any way, even commercially, with proper attribution. For these reasons, we cannot publish previously copyrighted maps or satellite images created using proprietary data, such as Google software (Google Maps, Street View, and Earth). For more information, see our copyright guidelines: http://journals.plos.org/plosone/s/licenses-and-copyright.

2.1.    You may seek permission from the original copyright holder of Figures 3 and 4 to publish the content specifically under the CC BY 4.0 license. 

2.2.    If you are unable to obtain permission from the original copyright holder to publish these figures under the CC BY 4.0 license or if the copyright holder’s requirements are incompatible with the CC BY 4.0 license, please either i) remove the figure or ii) supply a replacement figure that complies with the CC BY 4.0 license. Please check copyright information on all replacement figures and update the figure caption with source information. If applicable, please specify in the figure caption text when a figure is similar but not identical to the original image and is therefore for illustrative purposes only.

Reviewers' comments:

Reviewer's Responses to Questions

**Comments to the Author**

1. Is the manuscript technically sound, and do the data support the conclusions?

Reviewer #1: Yes

Reviewer #2: Yes

2. Has the statistical analysis been performed appropriately and rigorously? 

Reviewer #1: Yes

Reviewer #2: Yes

3. Have the authors made all data underlying the findings in their manuscript fully available?

Reviewer #1: Yes

Reviewer #2: Yes

4. Is the manuscript presented in an intelligible fashion and written in standard English?

Reviewer #1: Yes

Reviewer #2: Yes

5. Review Comments to the Author

Reviewer #1: Well written manuscript. I would recommend to include also the basic hematological and plasma biochemical profile of the turtles (to really understand their health status - this is standard procedure in clinical research, as well as the control of faeces for any kind of parasites).

Reviewer #2: This is a very unique and important study. It is very thorough and the statistical analyses have been performed adequately. Much useful information is presented to better understand turtle physiology. I have made a number of comments within the document: the biggest changes need to be in regards to re-working some of the figures, reference intervals by life-stage class, and showing some of the electrophoretograms for better interpretation of fractions. However, the majority of my comments are minor.

6. PLOS authors have the option to publish the peer review history of their article (what does this mean?). If published, this will include your full peer review and any attached files.

Reviewer #1: **Yes: **Prof. Dr. Zdeněk Knotek

Reviewer #2: No

---

## [Author Response · Author response to Decision Letter 0]

20 Sep 2021

We have responded to all of the reviewer comments in the "Response to Reviewer Comments" attachment.

---

## [Decision Letter · Decision Letter 1]

27 Sep 2021

Plasma electrophoresis profiles of Blanding’s turtles (Emydoidea blandingii) and influences of month, age, sex, health status, and location

PONE-D-21-17850R1

Dear Dr. Andersson,

We’re pleased to inform you that your manuscript has been judged scientifically suitable for publication and will be formally accepted for publication once it meets all outstanding technical requirements.

Kind regards,

Christopher M. Somers

Academic Editor

PLOS ONE

Additional Editor Comments (optional):

Reviewers' comments:

Reviewer's Responses to Questions

**Comments to the Author**

1. If the authors have adequately addressed your comments raised in a previous round of review and you feel that this manuscript is now acceptable for publication, you may indicate that here to bypass the “Comments to the Author” section, enter your conflict of interest statement in the “Confidential to Editor” section, and submit your "Accept" recommendation.

Reviewer #2: All comments have been addressed

2. Is the manuscript technically sound, and do the data support the conclusions?

Reviewer #2: Yes

3. Has the statistical analysis been performed appropriately and rigorously? 

Reviewer #2: Yes

4. Have the authors made all data underlying the findings in their manuscript fully available?

Reviewer #2: Yes

5. Is the manuscript presented in an intelligible fashion and written in standard English?

Reviewer #2: Yes

6. Review Comments to the Author

Reviewer #2: This really is an excellent paper. It shows us how both environmental, spatiotemporal, and biological factors influence plasma proteins in reptiles. All of my comments were addressed and the paper is now much clearer. Thank you for adding in the limitations section. It will be extremely useful for other individuals assessing health. I agree with the fractions and how they were cut as well. I have attached the document with some very minor suggestions (mostly grammatical). These can be done at the proof stage.

7. PLOS authors have the option to publish the peer review history of their article (what does this mean?). If published, this will include your full peer review and any attached files.

Reviewer #2: No

---

## [Editor Report · Acceptance letter]

7 Oct 2021

PONE-D-21-17850R1 

Plasma electrophoresis profiles of Blanding’s turtles (*Emydoidea blandingii*) and influences of month, age, sex, health status, and location 

Dear Dr. Andersson:

I'm pleased to inform you that your manuscript has been deemed suitable for publication in PLOS ONE. Congratulations! Your manuscript is now with our production department. 

Kind regards, 

on behalf of

Dr. Christopher M. Somers 

Academic Editor

PLOS ONE